# Preparation and Evaluation Mechanic Damping Properties of Fused Silica Powder@Polyurethane Urea/Cement Composites

**DOI:** 10.3390/ma15144827

**Published:** 2022-07-11

**Authors:** Hao Cheng, Peihui Yan, Fei Wan, Chao Feng, Yunfei Zhu, Ping Lv, Mingliang Ma

**Affiliations:** 1School of Civil Engineering, Qingdao University of Technology, Qingdao 266033, China; 15866623753@163.com (H.C.); yph2113155@163.com (P.Y.); shuoyiecool@sina.com (F.W.); zyf13165046056@126.com (Y.Z.); lvping-qd@163.com (P.L.); mamingliang@qut.edu.cn (M.M.); 2David International Design Institute of Shandong, Jinan 250014, China

**Keywords:** core-shell structure, constrained damping, polyurethane urea, cement, composite

## Abstract

In this paper, cement based on fused silica powder @ polyurethane urea (FSP@PUU) with a micro constrained damping structure was studied. Firstly, FSP@PUU core-shell particles were prepared by heterogeneous stepwise addition polymerization method and added into cement paste as damping filler to form a micro-constrained damping structure inside cement paste. The mechanical property and damping performance of cement-based composites were characterized by compressive strength, dynamic mechanical analysis (DMA) test and modal vibration test. The results showed that the damping performance of FSP @ PUU cement-based composites was affected by temperature, and the loss tangent of cement with 6wt% FSP@PUU increased to about 0.057 at −35 °C to 35 °C, which was 1.5 times cement paste within the glass transition temperature. With 6 wt% FSP@PUU, the damping ratio of cement-based composites increased by 58% compared with cement paste in the frequency range of 175–300 Hz, while the compressive strength decreased by only 5%. The cement with suitable FSP@PUU possesses excellent damping performance.

## 1. Introduction

Cement-based composites are one of the most widely used materials in building construction, bridges, rail transit and other projects [1,2,3]. Building structure design tends to be more and more “ultra-high rise, large span, complex”. These building structures in the process of service are more vulnerable to adverse dynamic loads in the environment (earthquake load, impact load, etc.), resulting in structural damage and reducing service life [4,5,6]. Therefore, it is very important to solve the vibration problem of structures. Structural vibration control can be divided into active control represented by installing dampers and passive control using vibration damping materials (polymers, graphene, fibers, etc.) to prepare cement-based damping composites. Pan et al. [5] developed a user-defined three-dimensional model to study the dynamic flexural responses of styrene–butadiene latex admixed concrete in the loss tangent, storage modulus, and loss modulus. When the 20% of styrene–butadiene latex was used, by weight of cement, it was found to favorably enhance the storage and loss moduli and the loss tangent of concrete. Ali et al. [7] evaluated the mechanical and dynamic properties of coconut fiber reinforced concrete (CFRC); the CFRC with higher fiber content has higher damping but lower dynamic and static modulus of elasticity. A fiber length of 5 cm and a fiber content of 5% has the best properties. Tian et al. [8] found that the damping ratio of specimens containing 70% damping aggregate (DA) was approximately three times higher than that of the reference mortar, with a slight decrease in the mechanical properties. Adding fiber was more effective than rubber powder in improving the damping ratio of the cement mortar, and the optimal dosage of fiber was 0.5%. In recent years, much research on the damping performance of cement-based composite has been published.

In polymer vibration damping of materials, the friction and movement of polymer chains can convert the mechanical energy of vibration noise into thermal energy consumption, which has a wide range of applications in the vibration damping of materials [9,10,11]. Polymer materials are added to the cement-based composite to increase the damping performance. Orak [12] investigated the damping characteristic when consisting of the same polyester resin ratio and different ratios of filler (quartz). It was observed that the critical damping ratio of polymer concrete was approximately four to seven times higher than that of cast iron. However, it has not been possible to determine if the damping characteristic of polymer concrete changes depends on the composition of the filler. Deredas et al. [13] studied the influence of doping polymer concrete with styrene–butadiene rubber (SBR) on its dynamic and mechanical properties. The results indicated that the sample with 30% of SBR had the lowest vibration amplitude value, while samples with 20% and 30% SBR had the highest values for the damping ratio. However, the addition of SBR to polymer concrete results in a decrease in its mechanical properties. Ahn et al. [14] proved that wave-type polymer concrete structure was embedded into the cement concrete sleeper, which can promote radiated rolling noise from rail track decreased about 4.22 dB by use of the polymer concrete embedment. Lee et al. [15] investigated the damping ratio of composites composed of preplaced aggregates and polyurethane matrix; the mental results proved that the damping ratio of polyurethane concrete was 12.7 times of ordinary concrete, while its compressive strength was decreased by 80%. This may be attributed to the fact that there is no bonding areas exist between the coarse aggregate in the polyurethane based composites. Xue et al. [16] proved that the damping ratio of rubberized concrete with 15 vt% rubber crumb was 62% higher than ordinary concrete, while the compressive strength decreased by about 45%, and the result showed that the addition of silica powder could slightly improve mechanical properties. Cao et al. [17] proposed a method of increasing the damping ratio of concrete by adding carboxylic styrene–butadiene latex. The damping ratio increased by 200%, while the compressive strength decreased by 18.3%, with a polymer cement ratio of 15wt%. In addition, viscoelastic damping materials were also used to study the vibration damping performance of cement-based composites. Lu et al. [18] investigated the damping characteristics of polyurea viscoelastic interlayer on concrete beams with constrained damping structure, and the results showed that the loss factor of interlayer concrete beams was improved by 3–7 times than concrete beams without damping. Huang et al. [19] used self-developed Qtech510 viscoelastic damping material with precast concrete slabs to form a restrained damping structure. The loss factor of concrete beams with a restrained damping structure could increase to more than ten times than the concrete beams without damping structure.

In this work, core-shell particles of fused silica powder @ polyurethane urea (FSP @ PUU) were prepared by heterogeneous stepwise addition polymerization using fused silica powder (FSP) as the base and viscoelastic damping material, polyurethane urea (PUU), as the damping layer. The core-shell particles were added into cement paste as damping filler to form the inside micro-constrained damping structure. The effects of different FSP and FSP @ PUU dosages on compressive strength and damping performance of cement-based composites were studied. Finally, the mechanism of cement-based composites’ performance change was clarified by combining microscopic morphology with macroscopic performance change in cement-based composites.

## 2. Experiment

### 2.1. Materials

P.O42.5 Portland cement was obtained from Shandong Shanshui Cement Group (Jinan, China). Fused silica powder (FSP) was obtained from Dezhou Jinghuo Technical Glass Co., Ltd. (Dezhou, China). The chemical composition of cement and FSP are listed in Table 1. The sample was prepared by the powder press method (mixed boric acid) for the test sample, and the data were measured using a 4 kW, 30 μm thin-window X-ray tube (Rigaku ZSX PrimusII X-ray fluorescence spectrometer). Qtech413-A (component A) and Qtech413-B (component B) viscous liquids were used to prepare polyurethane urea (PUU), component A was composed of semi-prepolymers such as diphenylmethane diisocyanate and toluene diisocyanate, and component B composed of diamino polyether, dihydroxy polyether, diamine chain extender, additives, etc. The mass ratio of the A component: B component was 1.1:1. They were obtained from Qingdao Shamu New Materials Co., Ltd. (Qingdao, China). Cyclohexane was purchased from Tian Beichen Founder Chemical Reagent Co., LTD. (Tianjin, China). Absolute alcohol and acetone were purchased from Sinopharm Chemical Reagent Co., Ltd. (Beijing, China). Water-reducing agents (PCA-I Polycarboxtlic acid superplasticizer) were obtained from Jiangsu Subote New Materials Co., LTD. (Nanjing, China). All the chemical reagents were analytically pure.

### 2.2. Micro Structure Characterizations

The characteristic vibrations of specimens were examined using Perkin–Elmer Fourier Transform Infrared spectrometer in the wave number range of 4000 cm^−1^–400 cm^−1^. The surface morphology and chemical compositions of the specimens were studied by Field Emission Scanning Electron Microscope (FE-SEM:Hitachi 3000-N) with Energy Dispersive X-ray Spectroscopy studies. Thermo Gravimetric Analysis (TGA) of the dehydrogenation process was carried out on a Netzsch TG209F3; an approximately 5 mg sample was loaded into an alumina crucible in the glove box using N_2_ as a carrier gas with the purge rate of 10 °C/min.

### 2.3. Mechanical Test

Twenty-millimeter cube specimens that were cured in the standard room (95% RH, 20 °C) for 28 days were used for testing cement-based composites’ compressive strength. The average of every three recording data was recorded, respectively.

The wdw-50kn electronic universal testing machine was used to calculate the compressive strength, and the formula is as follows:
(1)P=F/S

(*P* refers to the compressive strength, *F* refers to the force applied to the test block, and *S* refers to the area of the pressure action surface.)

### 2.4. Dynamic Thermomechanical Analysis Test

Thermal and mechanical properties were characterized using DMAT Q800 (TA Instruments USA, New Castle, DE, USA). The specimens were shaped in a prism of 4×12×60 mm and tested in a three-point bending mode. The tests were performed between −30 °C and −35 °C at a fixed frequency of 1 Hz, and the heating rate was 2 °C/min. Storage modulus $E^{\prime }$, loss modulusd $E^{\prime \prime }$ and loss tangent $tan\delta =E^{\prime \prime }/E^{\prime }$ were obtained [20].

### 2.5. Modal Vibration Test

The damping ratio of cement-based composites was investigated by modal vibration test of a cantilever beam using INV9810 impact hammer, INV9824 acceleration sensor and INV30262CLS10N multichannel data collection device, as shown in Figure 1. In order to induce vibration, a small impact hammer was used to produce small load impulses to the specimens. Firstly, the excitation force was applied to produce vibration using an impact hammer. Afterward, the vibration signal was imported into the system through the INV9824 acceleration sensor and INV30262CLS10N multichannel data collection device. Finally, the data were processed using the DASP-V11 modal analysis program. The data of 3 repeated hammer impacts were taken as the final analysis object. The amplitude and frequency curves of the test model were obtained by transfer function analysis of collected data. The damp ratio and modal frequency of the model were obtained by using the INV damping meter method.

### 2.6. Preparation of FSP@PUU and Cement-Based Composite

(1)Preparation of FSP@PUU

A 0.6 g component B/6.5 g FSP was dissolved in 80 mL cyclohexane at 75 °C under stirring, as shown in Figure 2a. After 15 min, 1 drop of the component A-acetone solution (50 mL acetone/0.66 g of component A) was added every 3 s. Then, the mixed solution was stirred for 2 h, as shown in Figure 2b. The suspension was filtered and washed three times with cyclohexane, anhydrous ethanol and deionized water to obtain the products. Lastly, the product was dried in a vacuum oven at 60 °C for 6 h and then ground in an agate mortar to obtain FSP@PUU particles with core-shell structure. The schematic synthesis FSP@PUU composite was shown in scheme Figure 3. In order to determine the optimum ratio of FSP@PUU, the following raw material formulations were selected for preparation, as shown in Table 2.

The schematic synthesis FSP@PUU composite was shown in scheme Figure 3. In this experiment, a non-homogeneous stepwise addition polymerization method was used. First, the component B with hydroxy FSP on the surface and PUU was mixed thoroughly in a cyclohexane solution, where the component B could be completely dissolved in the cyclohexane solvent. Meanwhile, the reactive group of component B was mainly composed of hydroxyl and amino groups, which were identical to the functional groups on the surface of FSP, and no chemical reaction would occur between them. The component A of PUU was insoluble in cyclohexane and was dispersed in the system as microdroplets. Secondly, due to the relatively large particle size of FSP, the probability of mutual impact with component A microdroplets was higher. This results in the diisocyanate group of component A reacting more readily with the hydroxyl group on the surface of the FSP. The hydroxyl groups on the surface of the FSP were converted into isocyanate groups as in reaction 1 of Figure 3. Then, the FSP with the isocyanate group could react with the component B in the cyclohexane solution again, and then the functional groups on the surface were reconverted to hydroxyl or amino groups as in the reaction 2 of Figure 3, and so on and so forth. Finally, a dense and continuous polyurea coating could be obtained on the FSP surface by in situ polymerization.

(2)Preparation of cement-based composite

The cement-based composite specimens were prepared by mixing cement and FSP(FSP@PUU). Cement and FSP(FSP@PUU) were mixed for 60 s. Then, water was added gradually in the following 120 s. The mixture was then added to prism and cube molds to form cement-based composite specimens, and all the samples were placed in the curing room (20 °C, 95% humidity) for 24 h. Then, the molds were removed, and the specimens were cured for 28 days. Static mechanical tests and vibration tests were carried out at the age of 28 days.

## 3. Result and Discussion

### 3.1. FTIR

Figure 4 shows FTIR spectra of the FSP and FSP@PUU in the 500–4000 cm^−1^ zone. In the FTIR spectra of FSP, the strong absorption bands at 3405 cm^−1^, 1105 cm^−1^ and 2970–2869 cm^−1^ were due to the stretching vibrations of −OH groups, Si-O-Si groups and CH_2_ and CH_3_ groups, respectively [21,22,23]. In the FTIR spectra of FSP @PUU, new peaks of N-H (3349 cm^−1^) bonds, C = O (1678 cm^−1^) groups and C-O-C(1105 cm^−1^) groups were found, and the peak at 3405 cm^−1^ of -OH disappeared [24]. In addition, the free NCO groups (2270 cm^−1^) vanished, which indicated that the reaction was completed [25]. All of them proved that FSP@PUU was successfully synthesized.

### 3.2. Surface Morphology Analysis

The morphologies images of FSP with different weights of Component A and B are shown in Figure 5. It was clarified that the coating of FSP particle’s surface was realized. With an increase in Component A and B content, the particles adhered together gradually, and the dispersity decreased, as shown in Figure 5a,d,g,j,m. FSP@PUU-1 was more dispersed while the coating layer was incomplete. PUU in FSP@PUU-3,4 was obviously excessive, and FSP@PUU particles were obviously stuck together. FSP@PUU-2 showed that individual particle coating with a smooth surface and good dispersion was achieved. The mean particle size of FSP@PUU-2 increased by 83% compared to FSP. Therefore, FSP@PUU-2 was selected to prepare cement-based composites. To further investigate whether the FSP surface is a PUU material, the elemental com-positions of FSP, PUU and cement were combined and elemental scans of C, N, O and Si were performed using EDX in the red area of Figure 6c, f and white area of Figure 6. FSP and FSP@PUU surfaces were inspected by elemental mapping analysis, as shown in Figure 6. It could be seen that the content of the C-element in FSP@PUU-1 was significantly higher than in FSP. From Figure 6d–f, it could be found that the content of element C in this white area was higher than elements O and Si. A comparison in Figure 6f showed that this area was covered with two layers of coating, and we marked it with numbers 1,2 in the Figure 5f. It significantly indicated the PUU coating on FSP, which confirmed the successful preparation of FSP@PUU composites.

### 3.3. TG-DTA

TG and DTG thermograms of FSP, PUU and FSP@PUU are shown in Figure 7. The curve of FSP had no weight loss at 10–800 °C. It was obvious that PUU and FSP@PUU had a similar tendency to undergo mass loss at 10–800 °C. According to the DTA curves of PUU and FSP@PUU, two main stages of thermal decomposition occurred. The first thermal decomposition stage was 175–350 °C, and the mass losses of PUU and FSP@PUU were 24.22% and 12.63%, respectively. FSP material had no mass loss at this stage, so the mass loss resulted from the decomposition of urethane and urea in the hard segment of PUU [26]. The second thermal decomposition stage was 350–450 °C, and the mass losses of PUU and FSP@PUU were 61.07% and 10.05%, respectively. The mass loss of FSP@PUU resulted from the thermal decomposition of polyether and other macromolecules in PUU [27]. In this study, compared with PUU, FSP@PUU exhibited higher thermal stability; the temperature of maximum thermal degradation rate increased from 326.65 °C to 332.50 °C in the first stage and from 383.90 °C to 397.87 °C in the second stage. The reason was that FSP had better heat resistance, and the −OH groups on FSP were chemically bonded with −NCO groups to coat the FSP surface, which required a higher degradation temperature [21,28].

### 3.4. Mechanical Properties

Figure 8 shows the variation compressive strength curves of cement with (2 wt%, 4 wt%, 6 wt%, 8 wt%, 10 wt%) FSP and FSP@PUU. Figure 9 shows that the compressive strength of cement-based composites increased by 0.54%, 1.4%, 6.6%, 8.7% and 14% with the increase in FSP. The reason is that FSP filled the pores as aggregate, and the interior of cement was more intensive, as shown in Figure 10a,b. The compressive strength of cement-based composites decreased by 0.50%, 4%, 5.8%, 5% and 13.7% with the increase in FSP@PUU doping amount. With more than 8wt% content of FSP@PUU, the interfacial adhesion decreases, and the compression limit value is easily reached under pressure [16,29,30,31]. In addition, PUU as polymer has a much lower elastic modulus than cement paste. The elastic modulus of cement-based composites would reduce by the addition of FSP@PUU. The compressive strength decreases with the increase in cube deformation under axial pressure [15,32].

### 3.5. DMA Test

Four doping amounts (2 wt%, 4 wt%, 6 wt% and 8 wt%) were selected according to compressive strength test results for damping performance experiments. Figure 9 shows the loss tangent changes with temperature at the same frequencies for the cement with different contents of FSP and FSP@PUU. FSP@PUU/cement composites exhibited the highest loss tangent among all the pastes, and one obvious peak occurred in the temperature range from −10 °C to 30 °C. The peak was regarded as the sign of glass transition, which was caused by the molecule chains’ movement of the FSP@PUU; as a result, more energy was consumed through molecular friction, dissipating as heat [33,34]. It should be noted that the loss tangent of cement with 6 wt% FSP@PUU increased to about 0.057 at −35 °C to 35 °C, which was 1.5 times of cement paste during the same testing temperature range. The result was that both FSP and FSP@PUU could further improve the damping capacity of cement. FSP@PUU showed much higher damping enhancing capacity than FSP in cement. For instance, the loss tangent of cement with 6 wt% FSP@PUU was 25% higher than the cement with 6 wt% FSP. In summary, the addition of viscoelastic material PUU could improve the damping performance of cement, which was best in the range of glass transition temperature.

### 3.6. Modal Vibration Test Results

#### 3.6.1. Amplitude and Frequency Curve Analysis

The vibration test of cantilever beam specimens was carried out, as shown in Figure 1. The amplitude–frequency curves of cement with different contents FSP and FSP @ PUU are shown in Figure 11. The vibration acceleration spectrum of FSP/cement composites showed that the peak vibration response decreased. The amplitude of cement with 6 wt% FSP decreased most significantly; the peak vibration response decreased by 36% and 24% compared with cement paste at the positions of first-order and second-order modal frequency, respectively. The vibration acceleration spectrum of FSP@PUU/cement composites showed that the peak vibration response decreased significantly at each order. At the first-order modal frequency, the vibration response peak of specimens with 6 wt% FSP@PUU decreased most significantly, which was 52% lower compared with cement paste. At the second-order modal frequency, the vibration response peak of the sample with 8 wt% FSP@PUU decreased most significantly, which was 46% lower compared with cement paste. Therefore, both FSP and FSP@PUU could improve the damping performance of cement, and the effect of FSP@PUU was more obvious.

#### 3.6.2. Modal Frequency and Damping Ratio

The modal frequency and damping ratio data are the average value of three 25×25×260 mm cantilever beams with the same mixture ratio, which are illustrated in Table 3. FSP@PUU/cement composites showed a 3−16% lower resonant frequency than FSP/cement. The FSP/cement composites frequencies were over 1−1.2 times higher modal than FSP@PUU/cement composites and cement paste; the reason is that the inclusion of FSP particles resulted in the increase in the stiffness to mass ratio of the composites [15]. Compared with cement paste, the damping ratio of cement with 6 wt% FSP increased by 44% and 30% in the frequency range of 175−300 Hz for the first-order mode and 1000−1500 Hz for the second-order mode. This was mainly due to the mutual deformation and friction between FSP particles and cement under vibration conditions, which improved the energy dissipation of the cement-based composites. Compared with cement paste, the damping ratio of cement with 6 wt% FSP@PUU increased by 58% and 21% for the first-order mode. In summary, both FSP and FSP@PUU could improve the damping performance of cement; FSP@PUU was more effective in the frequency range of 175−300 Hz in the first-order mode.

The main reasons were as follows: (1) There was a viscoelastic layer PUU between FSP and cement paste, which formed a microscopic constrained damping structure as in Figure 10d. Under vibration conditions, FSP and cement paste had different deformation, which caused shear deformation of PUU. Then, the internal molecular chains of PUU dissipated the vibration energy into heat energy and other energies by friction, which acted for energy dissipation in the cement-based composites [8,18,35,36]. With more than 8 wt% doping amount, the compressive strength of FSP@PUU/cement composites decreased substantially, and the inside interfacial pressure decreased at the same time. Under external forces, FSP@PUU would slip, and the shear deformation amount occurring in the damping layer was reduced, which affected the energy dissipation of microscopic constrained damping structure, and occurred the reduction in the damping ratio [18,19]. (2) FSP@PUU introduced more interfaces in cement, the friction on interfaces dissipated energy during the vibration, and the weakened PUU interfaces also increased energy dissipation [4,36].

## 4. Conclusions

This study proposed a new composite composed of cement and FSP@PUU core-shell particles with high damping performance. FSP@PUU core-shell particles were successfully synthesized by heterogeneous stepwise addition polymerization. In order to investigate the properties of the proposed cement cement-based composites, a series of experimental tests of the strength, the DMA test and the modal vibration test were conducted, and the following conclusions can be drawn:
(1)FSP @ PUU core-shell particle structure was successfully produced by the heterogeneous stepwise addition polymerization method. The core-shell particles with better encapsulation and dispersion were prepared by adjusting the material ratio and improving the manufacturing process. The experimental result showed that the optimum mass ratio of FSP and A, B components for coating was 10:1:0.9;(2)At the same loading frequency, the loss factor of FSP @ PUU/cement composites was affected by temperature. In the range of glass transition temperature, PUU had the most obvious influence on the loss factor of cement-based composites. The loss tangent of the cement with 6% FSP@PUU was 25% higher than 6% FSP at 8 °C. The cement with 6% FSP@PUU was 1.5 times higher than the cement paste at 8 °C;(3)With 6 wt% content of FSP @ PUU, the damping ratio of cement-based composites increased by 58% compared with cement paste in the frequency range of 175−300 Hz, and the compressive strength decreased by only 5%. This was mainly due to the energy dissipation of the micro-constrained damping structure composed of FSP-PUU-cement paste, which improved the damping performance of FSP @ PUU/cement composites.

## Figures and Tables

**Figure 1 materials-15-04827-f001:**
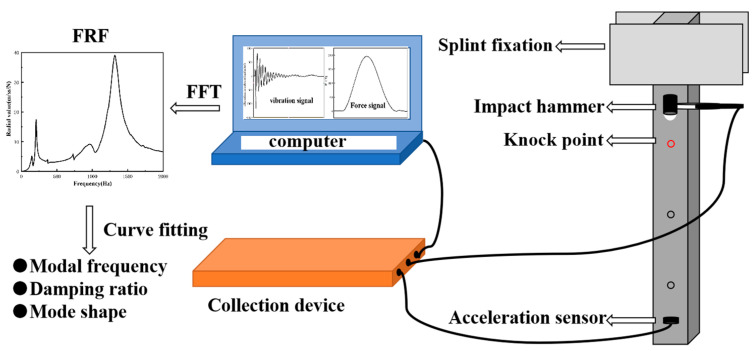
Experimental equipment of modal vibration test.

**Figure 2 materials-15-04827-f002:**
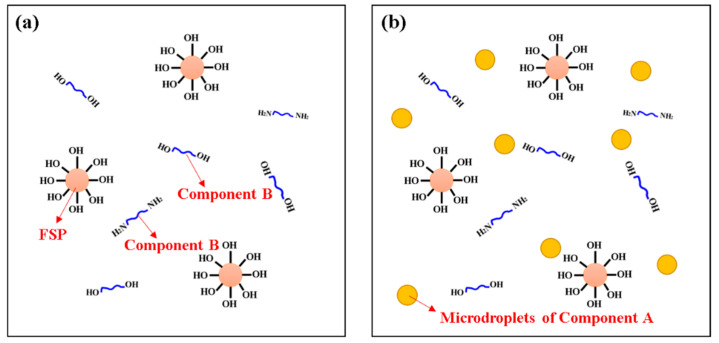
Component B and FSP reaction system (**a**); Component A, B and FSP reaction system (**b**).

**Figure 3 materials-15-04827-f003:**
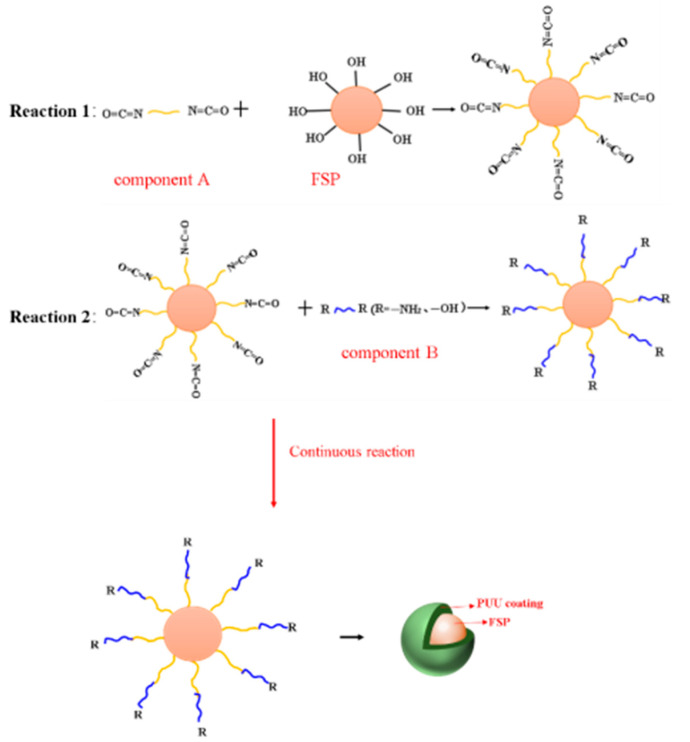
Schematic diagram of the preparation of FSP@PUU.

**Figure 4 materials-15-04827-f004:**
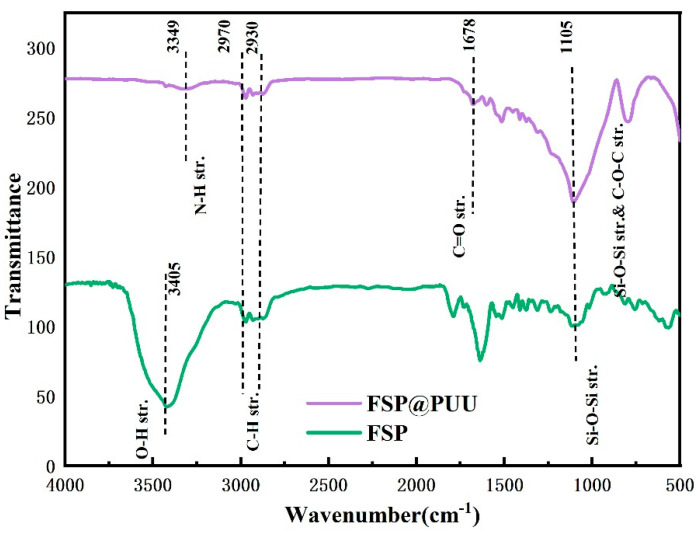
FTIR of FSP and FSP@PUU.

**Figure 5 materials-15-04827-f005:**
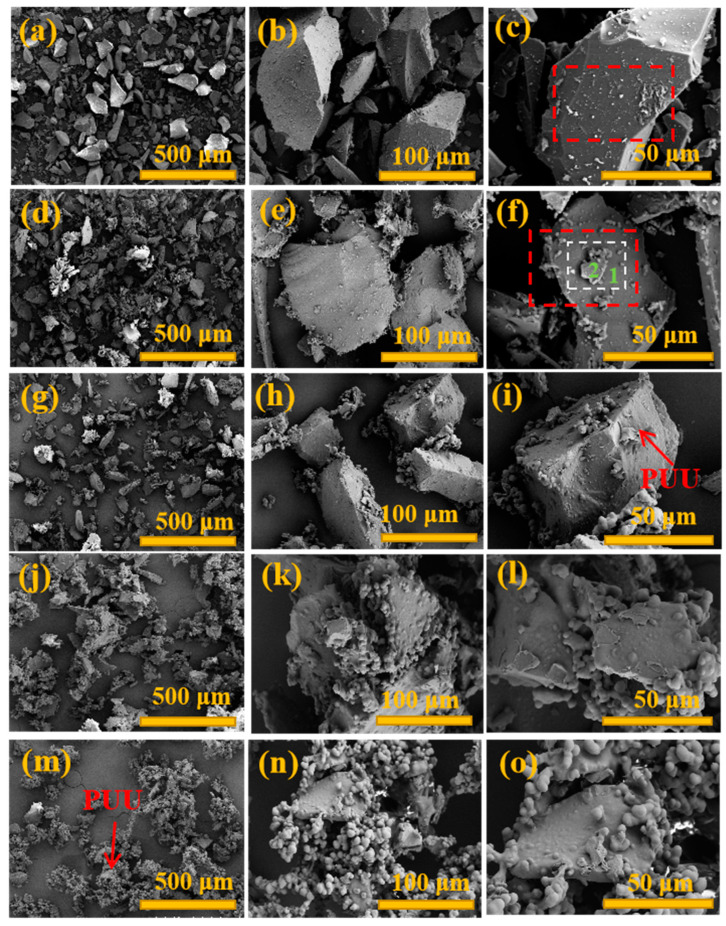
The SEM images of FSP (**a**–**c**); FSP@PUU-1 (**d**–**f**); FSP@PUU-2 (**g**–**i**); FSP@PUU-3 (**j**–**l**); FSP@PUU-4 (**m**–**o**).

**Figure 6 materials-15-04827-f006:**
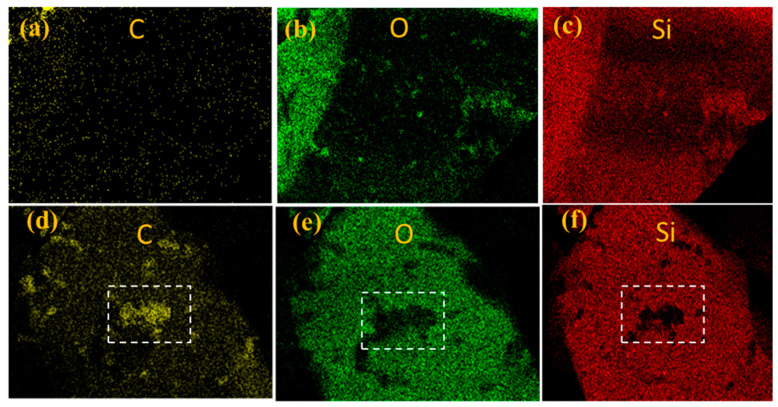
FSP (**a**–**c**) and FSP @ PUU-1 (**d**–**f**) corresponding to X-ray map of C, O and Si.

**Figure 7 materials-15-04827-f007:**
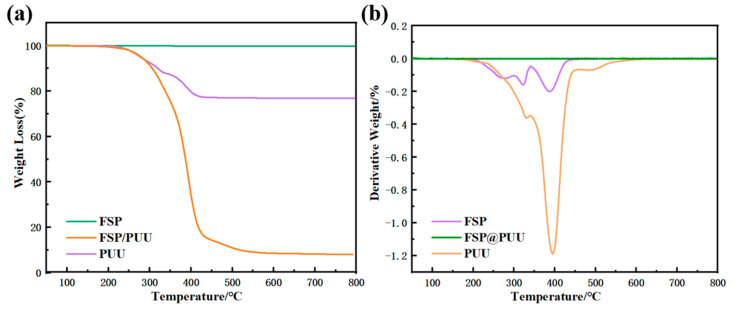
TGA (**a**) and DTG (**b**) curves of the FSP, PUU and FSP@PUU.

**Figure 8 materials-15-04827-f008:**
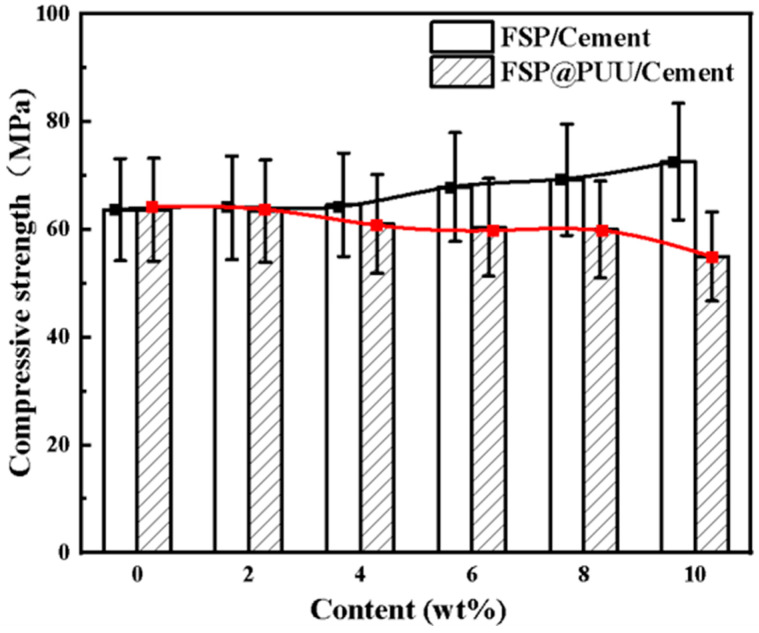
Compressive strength of 20 mm cube specimens for different FSP and FSP@PUU additions.

**Figure 9 materials-15-04827-f009:**
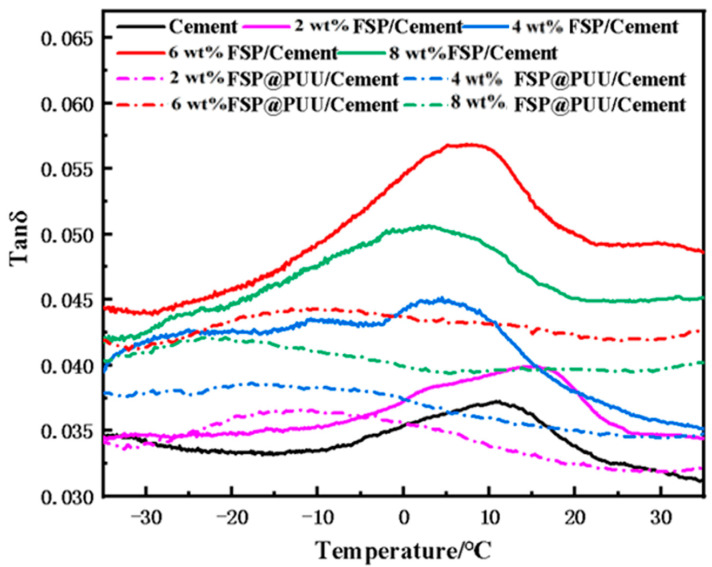
Loss tangent changes of 4×12×60 mm prismatic specimens for different FSP and FSP@PUU additions at different temperatures.

**Figure 10 materials-15-04827-f010:**
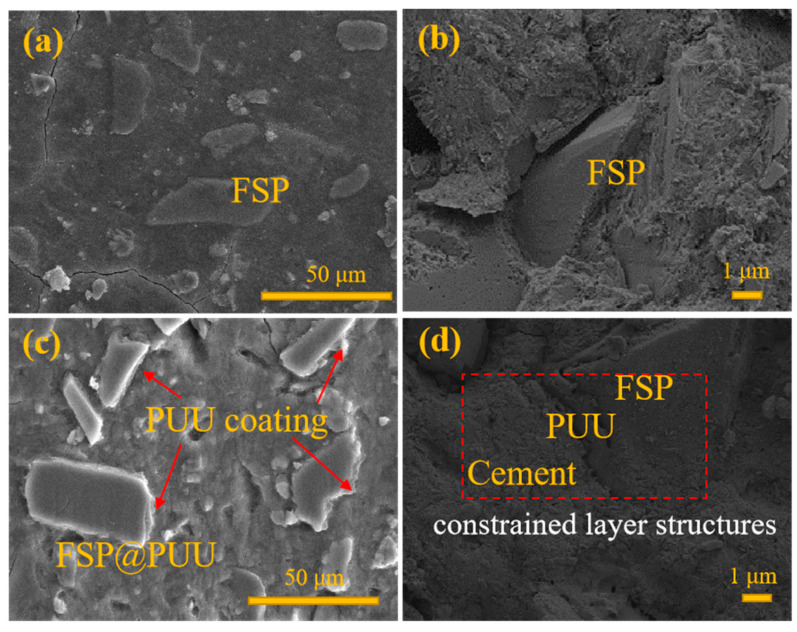
The SEM of cement (**a**,**b**) and FSP@PUU/Cement (**c**,**d**).

**Figure 11 materials-15-04827-f011:**
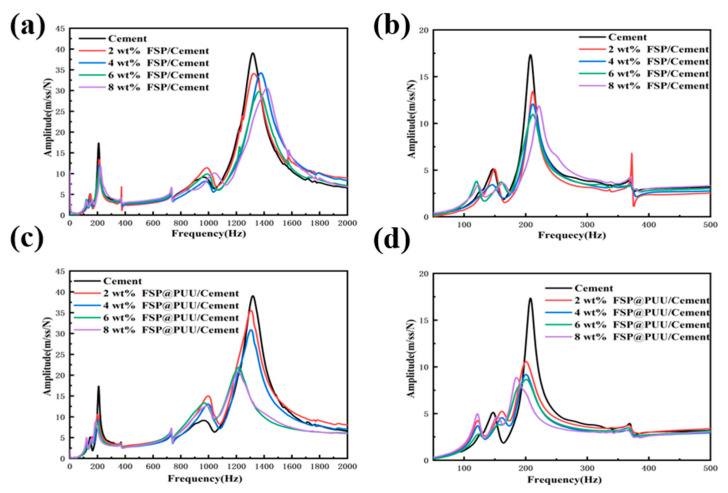
Vibration acceleration spectra of 25×25×260 mm prismatic specimens for different FSP (**a**,**b**) and FSP@PUU additions (**c**,**d**).

**Table 1 materials-15-04827-t001:** Chemical properties of cement and FSP.

Composition (wt.%)	SiO_2_	Al_2_O_3_	FexOx	CaO	MgO	SO_3_	Other
Cement	12.26	3.76	5.50	70.86	3.04	2.06	2.50
FSP	96.24	0.26	0.99	--	0.04	0.04	2.44

**Table 2 materials-15-04827-t002:** The raw material formulations for the preparation of FSP@PUU.

Sample	FSP (g)	Component A (g)	Component B (g)
FSP@PUU-1	6.50	1.71	1.55
FSP@PUU-2	6.50	1.10	1.00
FSP@PUU-3	6.50	0.66	0.60
FSP@PUU-4	6.50	0.56	0.51

**Table 3 materials-15-04827-t003:** Modal frequency and damping ratio of Cement, FSP/cement and FSP@PUU/cement.

Specimen	Modal Frequency (Hz)		Damping Ratio (%)(+−5%)	
	1nd	2nd	1nd	2nd
Cement	207.52	1318.36	3.59	4.61
2 wt%FSP/Cement	211.18	1324.46	3.75	5.43
4 wt%FSP/Cement	211.18	1324.46	3.97	5.55
6 wt%FSP/Cement	211.18	1362.30	6.40	6.63
8 wt%FSP/Cement	220.05	1418.46	4.83	6.49
2 wt%FSP@PUU/Cement	200.20	1304.93	6.60	4.93
4 wt%FSP@PUU/Cement	203.88	1321.41	7.68	5.00
6 wt%FSP@PUU/Cement	200.20	1220.70	8.50	5.83
8 wt%FSP@PUU/Cement	184.33	1220.70	8.33	6.11

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
