# Peer review of "Preparation and Evaluation Mechanic Damping Properties of Fused Silica Powder@Polyurethane Urea/Cement Composites"

_materials, 2022, doi:10.3390/ma15144827_

Round 1
Reviewer 1 Report
Paper entitled “Preparation and Properties of Fused Silica Powder@ Polyurethane Urea /Cement Composites” brings an interesting study about the FSP@PUU synthesis and application. Some suggestions for improvements can be seen in the attached document.

Author Response
The authors are grateful to reviewers’ rigorous attitude and constructive comments. This useful feedback would lead to significant improvements for this paper. We have considered these comments carefully and made corresponding revisions. A detailed list of responses to the editor’s and reviewers’ comments is provided as follows ,Please see the attachment.
However,due to my personal operation error, I uploaded three files before and saved them carelessly without finding the cancellation entry. The last two of the five files I want to submit to you. I am very sorry for my The names of the two files: response to reviewers1 and revised manuscript1.

Reviewer 2 Report
The manuscript entitled " Preparation and Properties of Fused Silica Powder Polyurethane Urea /Cement Composites " presents an interesting experimental study conducted on the obtaining and characterization of concrete with fused silica. However, the number of tested samples isn’t presented and many other issues must be addressed. The paper needs major revisions before it is processed further, some comments follow:
· The title is unclear. Please consider replacing the title with a clear formula that reflects the content of the manuscript.
· The introduction section could be improved. The citations have been introduced in the bulk -form [1-3] [4-6], [7-10], and not discussed separately. Please discuss the highlights individually and assure a clear correspondence between the affirmations from the manuscript and those from the cited papers.
· Also, please conduct a comprehensive and exhaustive study of the previous literature. Only six publications (considered between lines 37-59) are not enough. Please clearly highlight the pros and cons of previous results and justify the need for the current research.
· Table 1 - two types of iron oxides have been detected in these types of materials, therefore, please replace Fe2O3 with FexOy or provide the scientific proof to support your results. Moreover, which methods have been used to evaluate the properties presented in the table? Are these data obtained by the authors or they have been provided by the manufacturer (please introduce corresponding comments into the manuscript).
· "20mm cubes specimens" – Why this type of samples? Which standards have been used to evaluate the mechanical properties of the samples? (There is a reason in the standardized samples...). Please specify the standard requirements or the reason for choosing these values.
· DTG analysis. The curves show multiple peaks that haven’t been considered by the authors. Are those small peak errors? Please provide smoothed curves.
· FTIR and DTA analysis. How many samples have been tested? In the case of mixed materials (multiple components), there is a possibility that the inhomogeneities from the materials result in different behavior. Therefore, to clearly highlight the effect of different addition multiple samples should be tested in the same condition, then the mean values should be considered for comparison and discussion. As can be seen in Figure 10, the differences are very small, therefore, if the errors are considered, maybe the addition of FSP had no effect.
Author Response
The authors are grateful to reviewers’ rigorous attitude and constructive comments. This useful feedback would lead to significant improvements for this paper. We have considered these comments carefully and made corresponding revisions. A detailed list of responses to the editor’s and reviewers’ comments is provided as follows.Please see the attachmen.

Reviewer 3 Report
1. The keywords section needs to reorder in alphabetical order.
2. The significance of this study is not unclearly present, also the novelty of the present work is not solid that does not brings something really new regarding polyurethane urea /cement composites. Further, enhance on the introduction section is crucial to do.
3. Line 69 should be changed to “Materials and Methods”, not “Experiment”.
4. In section 2, it is needed to give additional illustrations regarding the workflow of present experiments to make them easier to understand rather than only using dominant text and specific figures.
5. The authors need to care regarding the uppercase and lowercase of the section and the subsection in the present manuscript should be revised.
6. The discussion section is not comprehensive, some detailed explanation of an aspect of mechanical properties is needed to extend.
7. Regarding previous comments, Polyurethane Urea along with cement composite have been potential applications in medical implants, especially total hip prostheses. The authors need to deliver this important point in the introduction and/or discussion section. Also, the suggested reference published by MDPI needs to be adopted to support this explanation as follows: Computational Contact Pressure Prediction of CoCrMo, SS 316L and Ti6Al4V Femoral Head against UHMWPE Acetabular Cup under Gait Cycle. J. Funct. Biomater. 2022, 13, 64. https://doi.org/10.3390/jfb13020064
8. Please revise the format and typesetting based on the MDPI format, it has several points that nor appropriate. Also, the authors can download the published version of Materials, MDPI, and compare it with the present manuscript to know the not appropriate things.
9. In figure 5, what is the crucial points that the authors want to deliver regarding the red box in this figure?
10. The authors need to compare the results with similar previous studies related.
11. Limitations in the present study should be mentioned.
12. Please rewrite the conclusion to make it into paragraphs, not point by point as present form.
13. Further research needs to be explained in the conclusion section.
Author Response
The authors are grateful to reviewers’ rigorous attitude and constructive comments. This useful feedback would lead to significant improvements for this paper. We have considered these comments carefully and made corresponding revisions. A detailed list of responses to the editor’s and reviewers’ comments is provided as follows,Please see the attachment.

Round 2
Reviewer 2 Report
The authors considered most of my comments and the manuscript was improved accordingly. Only one suggestion:
The citations in the bulk form should be removed. The authors should provide clear formulations related to the research presented in those articles and cite the papers in separate sentences.
Author Response
Thanks for your question. We're sorry to ignore to reflect the question in the manuscript. We have been modified as follows.
Pan et al[5] developed a user-defined three-dimensional model to study the dynamic flexural responses of styrene-butadiene latex admixed concrete in the loss tangent, storage modulus, and loss modulus. When the 20% usage of styrene-butadiene latex, by weight of cement, was found to favorably enhance the storage and loss moduli and the loss tangent of concrete. Ali et al.[7] evaluated the mechanical and dynamic properties of coconut fibre reinforced concrete (CFRC) ,the CFRC with higher fibre content has a higher damping but lower dynamic and static modulus of elasticity. When the fibre length of 5 cm and a fibre content of 5% has the best properties. Tian et al.[8] found that the damping ratio of specimens containing 70% damping aggregate (DA) was approximately three times higher than that of the reference mortar, with a slight decrease in the mechanical properties. Adding fiber was more effective than rubber powder in improving the damping ratio of the cement mortar, and the optimal dosage of fiber was 0.5% (Section 1 page 1-2)

Reviewer 3 Report
Good job, the authors have been addressed all of my comments.
Author Response
The authors are grateful to reviewers’ rigorous attitude and constructive comments. This useful feedback would lead to significant improvements for this paper. We have considered these comments carefully and made corresponding revisions.We have made some modifications on the basis of the original article and marked it with blue.Please see the attachment.We look forward to receiving your review report, thank you very much.
